# Job Mobility and Subjective Well-Being among New-Generation Migrant Workers in China: The Mediating Role of Interpersonal Trust

**DOI:** 10.3390/ijerph191811551

**Published:** 2022-09-14

**Authors:** Feng Zhang, Dan Liu, Xiaowei Geng

**Affiliations:** 1Jing Hengyi School of Education, Hangzhou Normal University, Hangzhou 311121, China; 2School of Educational Science, Ludong University, Yantai 264025, China; 3Institute for Education and Treatment of Problematic Youth, Ludong University, Yantai 264025, China

**Keywords:** new-generation migrant workers, subjective well-being, job mobility, interpersonal trust

## Abstract

New-generation migrant workers refers to those born in 1980 or thereafter, who become the majority of rural–urban migrants. New-generation migrant workers in Chinese cities are struggling with a lack of urban resources, which may lead to low well-being. On the basis of a questionnaire survey of 203 new-generation migrant workers, we used a multiple regression analysis to study new-generation migrant workers’ well-being and the mechanism underlying the effect of job mobility on well-being. The job mobility scale, interpersonal trust scale, and Affect Balance Scale were used. Results showed that job mobility was positively correlated with new-generation migrant workers’ subjective well-being and interpersonal trust, and interpersonal trust was positively correlated with subjective well-being. Interpersonal trust mediated the effect of job mobility on subjective well-being. In conclusion, job mobility can bring some benefits to new-generation migrant workers, that is, job mobility may increase their subjective well-being by increasing their interpersonal trust.

## 1. Introduction

Since the late 1980s, a large number of rural workers have been attracted from inland provinces to the southeast coastal region in China. Thus, the rural–urban migrate workers make up the largest group in workers of China. The number of rural migrant workers in China slightly decreased in 2020 due to the coronavirus pandemic (COVID-19), but soon rebounded again in 2021 and reached 292.5 million which comprises more than one third of the Chinese labor. Researchers have distinguished two generations of migrant workers, that is, the first-generation migrant workers who were born before 1980 and the new-generation migrant workers who were born in 1980 or thereafter (Qin and Huang, 2011) [1]. Due to the aging and retirement of first-generation migrant workers, new-generation migrant workers are becoming the majority of rural–urban migrants. New-generation migrant workers in Chinese cities are struggling with the shortage of urban resources, which may lead to low well-being. Thus, as the population of new-generation migrant workers increases, it is very important to pay attention to new-generation migrant workers’ subjective well-being.

Subjective well-being (SWB) includes people’s appraisals and evaluations of their own lives (Diener et al., 1999) [2]. It includes both reflective cognitive judgments such as life satisfaction as well as emotional responses to ongoing life in terms of positive and pleasant emotions versus unpleasant and negative emotions (Diener et al., 2018) [3]. He and Wang (2016) [4] found that, compared with first-generation migrant workers, new-generation migrants have a relatively higher level of subjective well-being. Many researchers studied the influential factors of new-generation migrant workers’ well-being, such as discrimination and family environment (Liu et al., 2013) [5], achievement motivation (Jin and Cui, 2013) [6], social identity (Zhang, 2014) [7], and job stress (Li et al., 2021) [8]. However, very few studies shed light on the effect of job mobility on subjective well-being of new-generation migrant workers.

With globalization and rapid economic growth in China, the issue of job mobility of migrant workers has received much attention. There are different definitions of job mobility, with one focused on the change in labor force status (i.e., movement between employment and unemployment and movement between employment and out of the labor force) and the other focused on the job-to-job transition of workers in different enterprises (Tian and Xu, 2015) [9]. Most previous studies on job mobility focused on the change in labor status and overlooked the importance of the movement of workers from one employer to another without a significant period of unemployment (Boon et al., 2008; Frazis and Ilg, 2009) [10,11]. According to Tian and Xu (2015) [9], in the present study, job mobility is defined as the intensity of the job-to-job transition of workers in different enterprises.

Compared with the first-generation migrant workers, new-generation migrants are better educated, more socially connected, change jobs more frequently, and obviously have higher job mobility than that of China’s urban resident workers (Tian and Xu, 2015; Zhou and Sun, 2010) [9,12]. Thus, it is important to study the effect of job mobility on new-generation migrant workers’ subjective well-being.

The present study aims to examine the effect of job mobility on new-generation migrant workers’ subjective well-being and its mediating mechanism, which may help to understand how job mobility influences subjective well-being among new-generation migrant workers, which is also useful for improving new-generation migrant workers’ subjective well-being.

### 1.1. Job Mobility and New-Generation Migrant Workers’ Subjective Well-Being

Previous research found that job mobility can bring many benefits. For example, job mobility is good for income, in which voluntary mobility (i.e., workers resign to start a new job) is much better than involuntary mobility (Groes et al., 2015; Lam et al., 2012; Light and McGarry, 1998) [13,14,15]. Recent evidence from the U.S. survey data reveals that many workers get paid more even after being forced to switch jobs (Chadi and Hetschko, 2021) [16]. Yankow (2022) [17] found that workers who demonstrate moderate job-changing in the first two years after labor market entry but then taper their mobility thereafter actually raise their wages most. Earning more money leads to greater subjective well-being. For example, drawing on 1,725,994 experience-sampling reports from 33,391 employed U.S. adults, Killingsworth (2021) [18] found that subjective well-being increased linearly with income, with an equally steep slope for higher earners as for lower earners.

In addition, job mobility could increase job satisfaction. Previous literature revealed that job satisfaction typically peaks initially following a job change but subsequently falls back to the baseline level over time (Boswell et al., 2005; Chadi and Hetschko, 2018) [19,20]. With 12,140 participants, Swaen et al. (2002) [21] found that changing jobs had a positive effect on employees with respect to job perception and job satisfaction. Specifically, before job changes, the mobility group reported significantly more conflicts with the supervisor, higher physical and emotional strain, higher degree of job insecurity, lower job satisfaction, and lower degree of commitment compared with employees who did not change jobs. After job changes, the mobility group reported improved autonomy, task diversity, decreased occurrence of conflicts with the supervisor, decreased physical and emotional strain, and improved training possibilities and job security than before the change. Chadi and Hetschko (2021) [16] analyzed the effects of job changes on subjective well-being using rich data from a representative German panel survey and found that job switchers report relatively higher levels of life satisfaction, at least after changing jobs for the first time. Therefore, we propose the hypothesis as follows:

**Hypothesis** **1:***Job mobility is positively correlated with new-generation migrant workers’ subjective well-being*.

### 1.2. Interpersonal Trust as the Mediator between Job Mobility and New-Generation Migrant Workers’ SWB

Trust is defined in terms of the intention to accept vulnerability based on positive expectations or beliefs regarding the interaction with other people in general (Rotter, 1967; Rousseau et al., 1998) [22,23]. Interpersonal trust is defined as an expectancy held by an individual that the word, promise, or statement of another individual or group can be relied on (Rotter, 1967) [22]. Previous studies on the relationship between job mobility and interpersonal trust focused on two aspects, i.e., trusted by others and trusting others. As for trusted by others, one previous study showed that job mobility may decrease people’s trust on the mover (Cao et al., 2020) [24]. However, few studies shed light on job mobility and the movers’ trust of others. According to the data from the General Social Survey, 1972–1994, Faught (2007) [25] revealed that job mobility types influenced the movers’ trust of others. In particular, downward job mobility was often correlated with lower levels of trust, with one exception: manual workers who move downward trust more than those who move upward. The higher job mobility may give new-generation migrant workers more opportunities to choose favorable relationships outside of one’s present committed relationships. Hence, we proposed that job mobility is positively related with new-generation migrant workers’ interpersonal trust.

**Hypothesis** **2:**
*Job mobility is positively correlated with new-generation migrant workers’ interpersonal trust.*


Previous studies showed that interpersonal trust is positively associated with happiness (Rodríguez-Pose and von Berlepsch, 2014) [26]. Poulin and Haase (2015) [27] found that trust increases well-being; higher trust predicted higher well-being. Bai et al. (2019) [28] found that trust both in acquaintances and strangers had a significant positive correlation with subjective well-being. Guo et al. (2022) [29] found that all types of trust positively predicted well-being at the individual level, and the effects of trust on well-being were enhanced by individualism. Hence, we propose that interpersonal trust is positively associated with new-generation migrant workers’ subjective well-being.

Based on Hypothesis 2, i.e., job mobility is positively correlated with new-generation migrant workers’ interpersonal trust, we could infer that interpersonal trust may play the mediating role in the effect of job mobility on well-being. Hence, we propose Hypothesis 3 as follows:

**Hypothesis** **3:***Interpersonal trust mediates the effect of job mobility on new-generation migrant workers’ subjective well-being*.

## 2. Method

### 2.1. Participants and Procedure

Using a convenience sampling method, we drew 203 new-generation migrant workers from several cities of east, middle, and west China; namely, Yantai and Jinan from east China, Taiyuan from middle China, and Xining from west China. The participants completed informed consent forms before testing. All procedures were carried out according to applicable ethics regulations. The final number of participants was 192 because 11 questionnaires were not completely answered. The final 192 participants comprised 112 males and 80 females with *M*_age_ = 2.19 (*SD* = 0.88). The sample characteristics were seen in Table 1.

### 2.2. Measures

#### 2.2.1. Subjective Well-Being

According to the definition of subjective well-being (Diener et al., 1999; 2018) [2,3], subjective well-being is not only the absence of negative mental states (such as worry, unhappiness, or loneliness), but also the experience of positive mental health. We measured new-generation migrant workers’ well-being using the Affect Balance Scale (Bradburn, 1969) [30], which includes five items concerned with the positive aspect of well-being, or ‘positive affect’, and five items concerned with the negative aspect of well-being, or ‘negative affect’. The scale focused respondents’ attention on experiences occurred ‘during the last few weeks.’ For example, items of positive affect included ‘particularly excited or interested in something’, ‘proud because someone had complimented you on something you had done’, and ‘pleased to have accomplished something’. Items of negative affect included ‘so restless that you couldn’t sit long in the chair’, ‘very lonely or remote from other people’, and ‘bored’. Participants were asked to answer whether they experienced these affects (Yes or No). Total scores for the positive affect and the negative affect scales were derived by summing the number of ‘Yes’ answers with the five items in each scale. Thus, a high positive affect score is considered to represent high psychological well-being and a high negative affect score is considered to represent low psychological well-being. The Affect Balance Score was derived by subtracting the negative affect score from the positive affect score, providing a scale ranging from −5 (minimum well-being) to +5 (maximum well-being).

#### 2.2.2. Job Mobility

In the present study, job mobility is defined as the intensity of job-to-job transition of workers in different enterprises (Tian and Xu, 2015) [9]. A variety of indicators are used to describe the job mobility of migrant workers, including the number of job changes and the proportion of second or multiple job changes (Bai and Li, 2008; Li, 1999, 2006; Tian and Yang, 2006; Xie, 1998; Xu, 2010) [31,32,33,34,35,36]. Based on the literature, job mobility was measured by asking new-generation migrant workers how many times they changed jobs since they left their rural hometown.

#### 2.2.3. Interpersonal Trust

Based on Rotter’s 25-item Interpersonal Trust Scale (Rotter, 1967) [22], Chun and Cambell (1974) [37] developed the short version of Rotter’s interpersonal trust scale, which consists of 12 items, i.e., the 3 marker items for each of 4 dimensions (interpersonal exploitation, political cynicism, societal hypocrisy, and reliable role-performance). *Interpersonal exploitation* concerns self-protection or caution based on a perception of others as exploitative and egocentric, for example, “In dealing with strangers one is better off to be cautious until they have provided evidence that they are trustworthy”, “It is safe to believe that in spite of what people say, most people are primarily interested in their own welfare”, and “In these competitive rimes one has to be alert or someone is likely to take advantage of you”. *Political cynicism* focuses on skepticism/cynicism about politicians and political bodies, for example, “This country has a dark future unless we can attract better people into politics”, “The judiciary is a place where we can all get unbiased treatment”, and “If we really knew what was going on in international politics, the public would have more reason to be frightened than they now seem to be”. *Societal hypocrisy* concerns hypocrisy in our society and the failure of impersonal, societal referents to meet commonly held expectations, for example, “Hypocrisy is on the increase in our society”, “Even though we have reports in newspapers, radio and television, it is hard to get objective accounts of public events”, and “Many major national sport contests are fixed in one way or another”. *Reliable role-performance* concerns hypocrisy and the failure to fulfill role requirements, for example, “Parents usually can be relied upon to keep their promises”, “Most salesmen are honest in describing their products”, and “Most repairmen will not overcharge even if they think you are ignorant of their speciality”. Participants were asked to rate their agreement with the items on 5-point scales (1 = Strongly disagree, 5 = Strongly agree). The higher the score, the higher the interpersonal trust.

### 2.3. Data Analysis

All questionnaire data were processed and analyzed by SPSS (the Statistical Package for the Social Sciences) which has been first developed by three PhD students at the University of Stanford and was sourced from Hangzhou, China, using statistical methods, such as Pearson correlation analysis, the independent sample *t*-test, ANOVA, and multiple hierarchical regression.

## 3. Results

### 3.1. Difference Analysis

Table 2 presents the means and standard deviations of all scales used in the present study. The results indicated that job mobility, interpersonal trust, positive emotions, negative emotions, and affect balance significantly differed according to gender, marital status, and age. As shown in Table 2, gender difference in the positive emotions reached a significant level (*p* < 0.05), where the positive emotions of females were significantly higher than that of males. The marital status difference in the job mobility reached a significant level (*p* < 0.001), where the job mobility of married new-generation migrant workers was higher than that of unmarried workers. The marital status difference in the affect balance also reached a significant level (*p* < 0.05), where the affect balance of married new-generation migrant workers was more positive than that of unmarried workers. The age difference in the job mobility, interpersonal trust, negative emotions, and positive emotions reached a significant level (*p* < 0.001). Specifically, the job mobility of people under 20 years old was significantly lower than that of the 20–30-year-olds (*p* < 0.05) and the 25–30-year-olds (*p* < 0.001). The positive emotion of those aged below 20 years old was significantly lower than that of 20–25-year-olds (*p* < 0.001) and 25–30-year-olds (*p* < 0.05). The negative emotion of those aged from 31 to 35 was significantly lower than that of those aged from 25 to 30 (*p* < 0.05) and from 31 to 35 (*p* < 0.05). The level of interpersonal trust in 31–35-year-olds was significantly lower than that in 25–30-year-olds (*p* < 0.05) and 31–35-year-olds (*p* < 0.05).

### 3.2. Descriptive Statistics and Correlation Analysis

As seen in Table 3, job mobility was positively correlated with new-generation migrant workers’ positive emotions (*p* < 0.001) and affect balance (*p* < 0.001), while negatively correlated with negative emotions (*p* > 0.05), which was consistent with Hypothesis 1. Job mobility was positively correlated with new-generation migrant workers’ interpersonal trust (*p* < 0.001), which was consistent with Hypothesis 2. Interpersonal trust was positively correlated with new-generation migrant workers’ positive emotion (*p* < 0.001) and affect balance (*p* < 0.05).

### 3.3. Mediation Analysis

On the basis of the correlation analysis, we found that job mobility was significantly correlated with positive emotions and affect balance, but not significantly correlated with negative emotions. Therefore, we tested the mediation of interpersonal trust in the effect of job mobility on positive emotions and affect balance.

According to Baron and Kenny (1986) [38], multiple hierarchical regression analysis was adopted to test the mediating effect of interpersonal trust between job mobility and new-generation migrant workers’ positive emotions, as seen in Table 4. In step 1, with positive emotions as the dependent variable and gender, marital status, age, length of time working away from hometown, and job mobility as predictors, Model 1 indicated that 10.1% of the variance of positive emotions could be attributed to gender, marital status, age, and length of time working away from hometown (*F* = 5.281, *p* < 0.001). Among them, gender had a significant negative predictive effect on positive emotions (*β* = −0.348, *p* < 0.05) and length of time working away from hometown had a significant positive predictive effect on positive emotions (*β* = 0.227, *p* < 0.05). Model 2 showed that job mobility had a significant positive predictive effect on positive emotions (*β* = 0.138, *p* < 0.05).

In step 2, with interpersonal trust as the dependent variable and gender, marital status, age, length of time working away from hometown, and job mobility as predictors, Model 3 indicated that 8.4% of the variance of interpersonal trust could be attributed to gender, marriage, age, and length of time working away from hometown (*F* = 4.268, *p* < 0.01). Among them, marriage had a significant negative predictive effect on interpersonal trust (*β* = −0.238, *p* < 0.01), and length of time working away from hometown had a significant positive predictive effect on interpersonal trust (*β* = 0.079, *p* < 0.01). Model 4 indicated that job mobility had a significant positive predictive effect on interpersonal trust (*β* = 0.042, *p* < 0.05).

In step 3, with positive emotions as the dependent variable and gender, marital status, age, length of time working away from hometown, job mobility, and interpersonal trust as predictors, Model 5 indicated that 8.7% of the variance of positive emotions could be attributed to job mobility and interpersonal trust (*F* = 7.120, *p* < 0.001). Interpersonal trust had a significant positive predictive effect on positive emotions (*β* = 0.789, *p* < 0.001), while job mobility did not significantly predict positive emotions (*β* = 0.105, *p* > 0.05). These results suggested that interpersonal trust mediated the effect of job mobility on new-generation workers’ positive emotions.

Further, according to Preacher and Hayes (2008) [39], we adopted the bootstrapping procedure to test the mediation of interpersonal trust between job mobility and positive emotions, with job mobility as the independent variable, interpersonal trust as the mediator, positive emotions as the dependent variable, and gender, marital status, age, and years of working outside as covariates. With a generating sample size of 5000, the bootstrapping procedure indicated that the indirect effect through the interpersonal trust was 0.033, which was significantly different from zero (95% CI = [0.0074, 0.0675]). It suggested that the effect of job mobility on new-generation migrant workers’ positive affect was mediated by interpersonal trust, see Figure 1.

As for affect balance, based on Baron and Kenny (1986) [38], multiple hierarchical regression analysis was performed to test the mediating role of interpersonal trust between job mobility and new-generation migrant workers’ affect balance, as shown in Table 5.

In step1, with affect balance as the dependent variable and gender, marital status, age, length of time working away from hometown, and job mobility as predictors, Model 1 indicated that 4.3% of the variance of affect balance could be attributed to gender, marital status, age, and length of time working away from hometown (*F* = 2.106, *p* > 0.05). Model 2 indicated that job mobility had a significant positive predictive effect on affect balance *(β* = 0.329, *p* < 0.001).

In step 2, with interpersonal trust as the dependent variable and gender, marital status, age, length of time working away from hometown, and job mobility as predictors, Model 3 indicated that 8.4% of the variance of interpersonal trust could be attributed to gender, marriage, age, and length of time working away from hometown (*F* = 4.268, *p* < 0.01). Among them, marital status had a significant negative predictive effect on interpersonal trust (*β* = −0.238, *p* < 0.01), and length of time working away from hometown had a significant positive predictive effect on interpersonal trust (*β* = 0.079, *p* < 0.01). Model 4 indicated that job mobility had a significant positive predictive effect on interpersonal trust (*β* = 0.042, *p* < 0.05).

In step 3, with affect balance as the dependent variable and gender, marital status, age, length of time working away from hometown, job mobility, and interpersonal trust as predictors, Model 5 indicated that 9.5% of the variance of affect balance could be attributed to job mobility and interpersonal trust (*F* = 4.952, *p* < 0.001). Job mobility had a significant positive predictive effect on affect balance (*β* = 0.300, *p* < 0.001) and interpersonal trust had a significant positive predictive effect on affect balance (*β* = 0.665, *p* < 0.05), which suggested that interpersonal trust partially mediated the effect of job mobility on affect balance.

Finally, according to Preacher and Hayes (2008) [39], we adopted the bootstrapping procedure to test the mediation of interpersonal trust between job mobility and affect balance, with job mobility as the independent variable, interpersonal trust as the mediator, affect balance as the dependent variable, and gender, marriage, age, and years of working outside as covariates. With a generating sample size of 5000, the bootstrapping procedure indicated that the indirect effect through the interpersonal trust was 0.028, which was significantly different from zero (95% CI = [0.0004, 0.0677]). It suggested interpersonal trust also mediated the effect of job mobility on new-generation migrant workers’ affect balance, see Figure 2.

## 4. Discussion

The present study aimed to explore the effect of job mobility on new-generation migrant workers’ subjective well-being and the mediating role of interpersonal trust.

### 4.1. Correlation of Job Mobility and New Generation Migrant Workers’ Subjective Well-Being

The present research showed that job mobility was positively correlated with new-generation migrants’ positive emotions and affect balance. Regression analysis showed that job mobility predicted new-generation migrants’ subjective well-being significantly after controlling for the effect of their gender, marital status, age, and length of time working away from hometown. This was consistent with the findings of Chadi and Hetschko (2021) [16], which analyzed the effects of job changes on subjective well-being using rich data from a representative German panel survey and found that job switchers reported relatively high levels of life satisfaction, at least after changing jobs for the first time. One possible explanation is that job mobility may bring new-generation migrant workers some benefits, for example, higher income (Groes et al., 2015; Lam et al., 2012; Light and McGarry, 1998) [13,14,15]. Even being forced to switch jobs, i.e., involuntary job changes, could lead to higher income (Chadi and Hetschko, 2021) [16]. Higher income usually brings higher experienced well-being [18]. Therefore, job mobility could bring new-generation workers’ increased well-being.

Previous literature showed that job insecurity, often associated with job mobility, is not associated with an increased subjective well-being; for example, Stankeviciūte et al. (2021) [40] found that job insecurity had a negative impact on employee subjective well-being. This study, different from that of Stankeviciūte et al., found that job mobility was positively related with new-generation migrants’ subjective well-being.

### 4.2. The Mediating Effect of Interpersonal Trust between the Job Mobility and New-Generation Migrant Workers’ Subjective Well-Being

Results of the mediating effect test showed that the effect of job mobility on new-generation migrant workers’ positive affect and affect balance was mediated by interpersonal trust. Higher job mobility leads people to greater interpersonal trust, which fosters cooperation among individuals, improves the level of perceived social support, and further increases well-being. This agrees with previous studies (Rodríguez-Pose and von Berlepsch, 2014) [26]. Poulin and Haase (2015) [27] found that trust increases well-being and higher trust predicted higher well-being. Bai et al. (2019) [28] found that both trust in acquaintances and trust in strangers had a significant positive correlation with subjective well-being. Guo et al. (2022) [29] found that all types of trust positively predicted well-being at the individual level, and the effects of trust on well-being were enhanced by individualism. Therefore, new-generation migrant workers’ interpersonal trust mediated the effect of job mobility on their subjective well-being.

### 4.3. Theoretical Implications and Practical Implications

Oishi (2014) [41] proposed socio-ecological psychology, which investigated how mind and behavior are shaped in part by their natural and social habitats. In this article, we specifically focused on one socio-ecological factor, i.e., job mobility. So far, very few studies have tested the effect of job mobility on new-generation migrant workers’ subjective well-being from a social ecology perspective. We found that job mobility, as a socio-ecological factor, not only directly influences new-generation migrant workers’ subjective well-being, but also influences it indirectly by interpersonal trust, which is an important contributor to this field.

The current study also has important implications. Enhancing new-generation migrant workers’ urban integration and city identification is a big challenge for China. This study can give suggestions on how to increase new-generation migrant workers’ well-being and further enhance their urban integration. For instance, increasing job mobility can enhance new-generation migrant workers’ well-being.

However, there were also limitations in the present research. First, job mobility may have different types, for example, voluntary or involuntary job mobility. Those who have bad experiences at previous workplaces are more likely to leave voluntarily, which may also influence their attitudes regarding interpersonal trust. Therefore, researchers should analyze the effect of different kinds of job mobility (i.e., voluntarily or involuntarily)on subjective well-being in the future. Second, in the present research, the sample size was not large and was obtained by a convenience sampling method. A larger sample obtained by a probability sampling method would be better in the future. These results were only preliminary and have to be confirmed by further larger and well-designed studies.

## 5. Conclusions

The present research found that job mobility was positively correlated with new-generation migrants’ well-being and interpersonal trust. Interpersonal trust mediated the effect of job mobility on new-generation migrants’ well-being. Job mobility can bring some benefits to new-generation migrants; that is, job mobility may increase their well-being by increasing their interpersonal trust. These results need to be confirmed by further studies.

## Figures and Tables

**Figure 1 ijerph-19-11551-f001:**
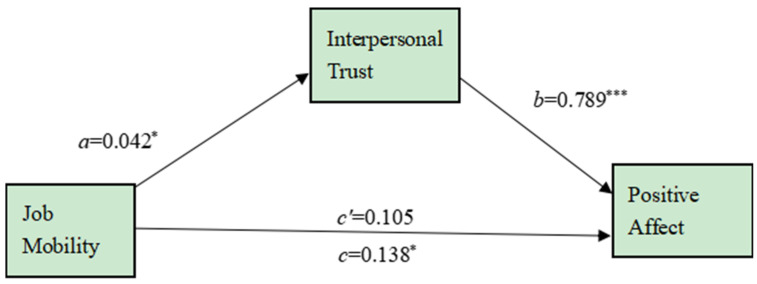
The mediating effect of interpersonal trust in the relationship between job mobility and positive affect. Note. * *p* < 0.05, *** *p* < 0.001.

**Figure 2 ijerph-19-11551-f002:**
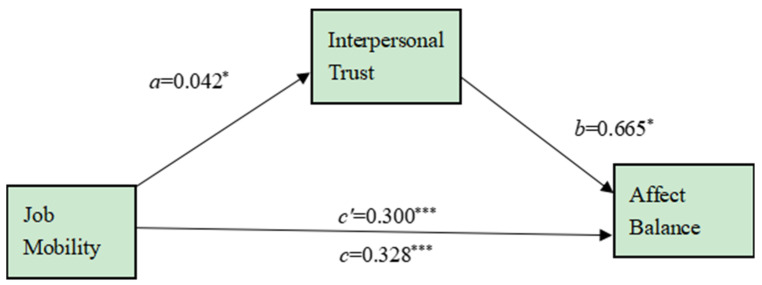
Beta coefficients for the mediating effect of interpersonal trust in the relationship between job mobility and affect balance. Note. * *p* < 0.05, *** *p* < 0.001.

**Table 1 ijerph-19-11551-t001:** Sample characteristics.

Measure.	Items	Sample Size (*n*)	Percentage (%)
Gender	Male	112	58.3
	Female	80	41.7
Marital Status	Yes	52	27.1
	No	140	72.9
Age	Below 20	44	22.9
	20–25	84	43.8
	26–30	48	25.0
	31–35	16	8.3
Education	Junior middle school	50	26.0
	High middle school	90	46.9
	Vocational school	37	19.3
	Undergraduate	15	7.8
Length of time working away from hometown	Below 1 year	55	28.6
	1–2 years	44	22.9
	3–5 years	57	29.7
	6–10 years	26	13.5
	Above 10 years	10	5.2
Monthly Income	Below Ұ1500	8	4.2
	Ұ1500–Ұ3000	102	53.1
	Ұ3000–Ұ5000	60	31.3
	Above Ұ5000	22	11.5

**Table 2 ijerph-19-11551-t002:** Difference analysis of the scale scores of the present study (*n* = 192).

Group	Statistic	Job Mobility	Interpersonal Trust	Positive Emotions	Negative Emotions	Affect Balance
Gender	Male	2.38 ± 1.66	3.27 ± 0.32	2.35 ± 1.16	3.43 ± 1.48	−1.08 ± 1.76
	Female	2.21 ± 1.56	3.28 ± 0.45	2.74 ± 1.10	3.54 ± 1.45	−0.80 ± 1.42
	*t*	0.762	−0.062	−2.342 *	−0.507	−1.179
	*p*	0.447	0.951	0.020	0.613	0.240
Marital status	Married	2.98 ± 1.42	3.21 ± 0.43	2.71 ± 0.89	3.19 ± 1.27	−0.48 ± 1.48
	Unmarried	2.07 ± 1.62	3.30 ± 0.36	2.44 ± 1.22	3.58 ± 1.52	−1.14 ± 1.65
	*t*	3.574 ***	−1.490	1.483	−1.631	2.544 *
	*p*	0.001	0.138	0.140	0.105	0.012
Age	Below 20	1.45 ± 1.11	3.19 ± 0.31	1.93 ± 1.17	3.09 ± 1.60	−1.16 ± 1.41
	20–25	2.13 ± 1.71	3.32 ± 0.33	2.64 ± 1.09	3.68 ± 1.47	−1.04 ± 1.74
	26–30	3.25 ± 1.42	3.21 ± 0.51	2.77 ± 1.19	3.69 ± 1.32	−0.92 ± 1.77
	31–35	2.88 ± 1.31	3.43 ± 0.16	2.63 ± 0.72	2.81 ± 1.11	−0.19 ± 0.75
	*F*	12.436 ***	2.718 *	5.306 **	3.073 *	1.508
	*p*	0.001	0.046	0.002	0.029	0.214

* *p* < 0.05, ** *p* < 0.01, *** *p* < 0.001.

**Table 3 ijerph-19-11551-t003:** Descriptive statistics and correlation matrix for all variables (*n* = 203).

	*M*	*SD*	1	2	3	4	5	6
1	2.438	1.187	1					
2	2.318	1.614	0.542 **	1				
3	2.510	1.149	0.273 **	0.271 **	1			
4	3.474	1.465	0.049	−0.122	0.244 **	1		
5	−0.964	1.626	0.149 *	0.301 **	0.486 **	−0.728 **	1	
6	3.273	0.377	0.180 *	0.225 **	0.314 **	0.050	0.177 *	1

* *p* < 0.05, ** *p* < 0.001. 1= length of time working away from hometown, 2 = job mobility, 3 = positive emotions, 4 = negative emotions, 5 = affect balance, and 6 = interpersonal trust.

**Table 4 ijerph-19-11551-t004:** Multivariate hierarchical regression analysis of the impact of interpersonal trust and job mobility on positive emotions (*n* = 192).

Independent Variable	Step1: Positive Emotions		Step 2: Interpersonal Trust		Step3: Positive Emotions
	Model 1 β	Model 2 β	Model 3 β	Model 4 β	Model 5 β
Gender	−0.348 *	−0.391 *	−0.002	−0.015	−0.379 *
Marital status	−0.181	−0.176	−0.238 **	−0.236 **	0.011
Age	0.120	0.111	0.040	0.038	0.082
Length of time working away from hometown	0.227 *	0.127	0.079 **	0.049	0.089
Job mobility		0.138 *		0.042 *	0.105
Interpersonal trust					0.789 ***
*R* ^2^	0.101	0.128	0.084	0.106	0.188
Adjusted *R*^2^	0.082	0.104	0.064	0.082	0.161
*F*	5.281 ***	5.450 ***	4.268 **	4.429 **	7.120 ***

* *p* < 0.05, ** *p* < 0.01, *** *p* < 0.001. Gender was converted into a dummy variable: 1 = male, 0 = female; Marital status was also converted into a dummy variable: 1 = married, 0 = unmarried.

**Table 5 ijerph-19-11551-t005:** Multivariate hierarchical regression analysis of the impact of interpersonal trust and job mobility on affect balance (*n* = 192).

Independent Variable	Step 1: Positive Emotions		Step 2: Interpersonal Trust		Step 3: Affect Balance
	Model 1 β	Model 2 β	Model 3 β	Model 4 β	Model 5 β
Gender	−0.226	−0.328	−0.002	−0.015	−0.318
Marital status	0.538	0.551	−0.238 **	−0.236 **	0.708 *
Age	−0.041	−0.062	0.040	0.038	−0.087
Length of time working away from hometown	0.123	−0.114	0.079 **	0.049	−0.146
Job mobility		0.328 ***		0.042 *	0.300 ***
Interpersonal trust					0.665 *
*R^2^*	0.043	0.117	0.084	0.106	0.138
Adjusted *R^2^*	0.023	0.093	0.064	0.082	0.110
*F*	2.106	4.937 ***	4.268 **	4.429 **	4.952 ***

* *p* < 0.05, ** *p* < 0.01, *** *p* < 0.001. Gender was converted into a dummy variable: 1 = male, 0 = female; Marital status was also converted into a dummy variable: 1 = married, 0 = unmarried.

## Data Availability

According to the data access policies, the data used to support the findings of this study are available upon a reasonable request made by email: fengandwei@126.com.

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
