# Peer review of "Job Mobility and Subjective Well-Being among New-Generation Migrant Workers in China: The Mediating Role of Interpersonal Trust"

_ijerph, 2022, doi:10.3390/ijerph191811551_

Round 1

Reviewer 1 Report

I am sorry to say, but in my opinion, the quality of the manuscript does not reach the level that can be expected at a Q1/Q2 level journal. I have several concerns with the manuscript at different points.

Most importantly, the sample. The authors use a 200-person convenience sample. No more information is disclosed about the sampling procedure. I believe researchers put serious efforts and resources to create probability samples for reason. Theoretically, they are necessary to make them be able to infer conclusions from the sample to the population, and in a strict sense, without a probability sample, statistical analysis is meaningless. The validity of the results will be limited to the sample only, which in this case is completely unknown.

With respect to theory, I find H1 contradicting the presented literature. Further, I find H3 unnecessary (and somewhat trivial) – to formulate H4, it could be assumed from previous literature.

About the statistical analysis, the authors test their main hypothesis (the mediating role of trust in the effect of job mobility on wellbeing) by comparing the coefficients in the job mobility when predicting well-being in the regressions where trust is included and when it is not. First, testing the difference of a parameter in two regressions is possible, to my knowledge the Wald test is a standard procedure for this. However, to test the mediating role of Z in the regression of X predicting Y, the standard method is a different one, estimating a regression with the interaction effect of X and Z:  Y=a+b_1*X+b_2*Z+b_3*X*Z.

Further, the examined variables are supposed to be correlated due to different directions of causality. Interpersonal trust may not only predict subjective well-being but it can be influenced by it by psychological rationalization. Job mobility may also be endogeneous: those, who have bad experiences at previous workplaces are more likely to leave voluntarily or involuntarily, which may also influence their attitudes regarding interpersonal trust. These mechanisms are not analyzed in the paper. 

Reviewer 2 Report

The manuscript is interesting, but the authors should make corrections related to the applied statistical methods.
(1) The sentence in lines 133-134 should be removed from the text: "After finishing the survey, the participants 133 received a small gift, valued about at US$1.5 (i.e., a bottle of hand soap). "
(2) In Table 1, column 3, the phrase "Frequency" is used. The data in this column refers to the number of subgroups. The heading should read "Sample size".
(3) The responses to the survey question are on a Likert scale. In this case, the classical measures that are the arithmetic mean and standard deviation should not be used. The median and the quarter deviation are better. And then the Mann-Whitney U test should be used.
(4) Also, if the arithmetic mean and standard deviation are used, the Student's t-test is incorrect. The t-test is used when the distribution is normal or similar. In this case the variable is measured on an ordinal scale and non-parametric tests based on ranks should be used e.g. Mann-Whitney U test.
(5) The authors used the Pearson product-moment correlation coefficient. In this case it is incorrect. For ordinal variables the Kendall's tau coefficient is appropriate.
(6) In regression analysis, ordinal variables should be converted into dichotomous dummy variables.

Round 2

Reviewer 2 Report

The author took into account the suggestions to the improved version of paper.

Congratulation for the work involved in this study!.

Author Response

Thank you very much for your comments.